# Unraveling Psychiatric Disorders through Neural Single-Cell Transcriptomics Approaches

**DOI:** 10.3390/genes14030771

**Published:** 2023-03-22

**Authors:** Samar N. Chehimi, Richard C. Crist, Benjamin C. Reiner

**Affiliations:** Department of Psychiatry, Perelman School of Medicine, University of Pennsylvania, Philadelphia, PA 19104, USA

**Keywords:** single-nuclei RNA-seq, transcriptome, psychiatric disorders, cellular characterization

## Abstract

The development of single-cell and single-nucleus transcriptome technologies is enabling the unraveling of the molecular and cellular heterogeneity of psychiatric disorders. The complexity of the brain and the relationships between different brain regions can be better understood through the classification of individual cell populations based on their molecular markers and transcriptomic features. Analysis of these unique cell types can explain their involvement in the pathology of psychiatric disorders. Recent studies in both human and animal models have emphasized the importance of transcriptome analysis of neuronal cells in psychiatric disorders but also revealed critical roles for non-neuronal cells, such as oligodendrocytes and microglia. In this review, we update current findings on the brain transcriptome and explore molecular studies addressing transcriptomic alterations identified in human and animal models in depression and stress, neurodegenerative disorders (Parkinson’s and Alzheimer’s disease), schizophrenia, opioid use disorder, and alcohol and psychostimulant abuse. We also comment on potential future directions in single-cell and single-nucleus studies.

## 1. Introduction

The pathophysiology of psychiatric disorders includes delicate dysregulation of cellular and molecular mechanisms. Transcriptomic studies have primarily relied on bulk RNA sequencing (RNA-seq) approaches, which have helped to characterize relevant brain regions and identify differentially expressed genes (DEGs) in the disease context [1]. However, bulk RNA-seq only provides an overall quantification of transcript expression and does not capture the heterogeneity of distinct expression patterns in different cell populations. While previous efforts have attempted to mitigate this limitation using fluorescence-activated cell sorting, laser capture microdissection, and computational methods to deconvolute and to enrich cell populations, these are laborious and require adequate marker selection for purification of the targeted cell population and precise downstream analysis of the composition of bulk data, respectively [2,3,4].

Neural cellular heterogeneity varies in the type and abundance of neuronal and non-neuronal populations by neuroanatomical region. Cellular heterogeneity, the neural connectome, disease onset, and neuronal plasticity of specific regions play important roles in psychiatric disease and substance use disorders. The development of single-cell and single-nucleus RNA sequencing (scRNA-seq and snRNA-seq, respectively) approaches have allowed for the deconvolution of cell type-specific expression signatures and characterization of the cellular diversity and heterogeneity in different tissues [2,3,5].

The approaches for single-cell transcriptomic profiling have evolved from the use of multiple barcode attachment rounds and multiplexing in plates to microfluidic systems capable of capturing individual cells in small, barcoded droplets and assigning each transcript a unique molecular identifier in a high-throughput fashion [5]. Consequently, it is possible to identify the genome-wide expression profile of neural cells that lead to the complex pathophysiological processes observed in substance use and psychiatric disorders [6,7].

The difficulty in obtaining fresh human brain tissue samples that allow for the collection of cells with acceptable quality presents a challenging hindrance to exploring single-cell gene expression. A more feasible approach to studying human brain cells is the use of isolated nuclei from postmortem brain tissue to characterize neuronal populations [3,7]. Although nuclei datasets might miss specific transcripts, such as dendritically transported transcripts in neurons, literature evidence supports snRNA-seq broadly recapitulating the results of single-cell studies [8,9]. In contrast to human samples, animal and cell culture models of psychiatric disease and substance use disorders present an opportunity for utilizing either single-cell or single-nucleus approaches, with approach selection frequently driven by experimental design [2,5].

Traditional scRNA-seq and snRNA-seq analysis workflows begin by performing quality control to remove low-quality or uninformative nuclei, eliminate putative multiplets, and reduce background noise [10]. While beyond the scope of this review, a variety of pipelines have been proposed to automate a consistent statistical approach to these quality control assessments, including multimodal analysis [10,11,12].

The expression of cell type-specific markers, cross-referenced with prior publications and current publicly available datasets, allows for the identification of known and novel cellular subtypes [3,7,13,14]. Recent efforts from large consortiums generated cellular and molecular profiling from the whole brain in multiple species [15,16,17] and resulted in the Azimuth reference dataset, which gathers maps for multiple available single-cell references, including brain data. The importance of integrating data from single-cell studies and available datasets is establishing patterns and creating better cell classification and data comparison standards for psychiatric disorders, as recently covered by Olislagers et al. (2021) [18].

Differential expression in identified cellular subtypes is then determined, with approaches varying in both the weighting applied to the importance of individual cells and nuclei [13,19,20]. To ensure accurate differential gene expression analysis, pseudo-bulk approaches can be used to aggregate individual cell counts to the sample level and overcome the challenges of intraindividual single-cell heterogeneity [21]. Downstream analyses frequently examine the overrepresentation of cellular subtype-specific differentially expressed genes in molecular signaling pathways. Data obtained from genome-wide association studies (GWAS) of psychiatric disease and substance use disorders can be combined with single-unit transcriptomics data to identify the cellular subtypes associated with GWAS loci [22,23].

Many psychiatric diseases are being investigated under single-cell resolution. For the scope of this review, we will not include neurodevelopmental diseases and will focus on depression and stress, neurodegenerative disorders (Parkinson’s and Alzheimer’s diseases), schizophrenia, and substance use disorders (opioids, alcohol, and psychostimulants). In this review, we explore the current findings on the brain transcriptome, addressing molecular alterations in neural cell type-specific transcriptomes identified in human and animal model studies in the beforementioned psychiatric disorders. We also comment on potential future directions in single-cell and single-nucleus studies in psychiatric diseases and substance use disorders.

## 2. Major Depressive Disorder and Stress

Major depressive disorder (MDD) is the most common psychiatric disease globally, characterized by a broad spectrum of clinical symptoms, such as alterations of mood that lead to behavioral changes with increased sadness, inability to experience pleasure and motivation, impairments in psychosocial function, diminished interest in most or all activities, and increased negative and suicidal thoughts [24,25,26].

One brain region explored in multiple molecular and cellular studies of MDD is the prefrontal cortex (PFC) due to its relevance in mediating cognitive and behavioral functions [26]. In the PFC of a mouse model of depression-like behavior caused by social isolation, decreased total numbers of NeuN+ neurons and increased Iba1+ microglia were observed by immunohistochemistry [27]. Inflammatory signals in the PFC of mouse models of stressful condition- or virally-induced blood-brain barrier (BBB) permeability had altered gene expression with the consequent manifestation of social avoidance behavior and mood alteration [28,29].

Previous bulk RNA-seq and small RNA-seq (miRNA expression) studies in humans with MDD have shown global neural changes in postmortem samples and highlighted downregulation in glutamatergic neuron and oligodendrocyte (OL) differentiation pathways [30,31]. Through immunohistochemistry, it was also observed decrease in *CLDN5* mRNA expression and changes in PFC endothelial cells in postmortem tissue collected from women diagnosed with MDD [28].

Further evidence of OL alterations came with snRNA-seq and pseudo-time trajectory strategies that showed that oligodendrocyte precursor cells (OPC) are a commonly affected cell type in the PFC during the development of the depressive state [32,33].

Single-nucleus studies of human transcriptome profiles from persons with MDD have revealed other considerable findings: decreased expression of myelin-related genes in the immune-oligodendrocytes cluster (oligodendrocyte-lineage characterized by immune-related properties) compared to mature OL [33]; upregulation of reactive oxidative species production, structural atrophy of OPCs followed by OL apoptosis, and expression of immune markers in oligodendrocyte-lineage cells [33]; and transcriptional dysregulation in the cortical layers, affecting mainly excitatory neurons located in deeper cortical layers, in the brains of males with MDD [32].

In addition to the PFC, hippocampi from mice exposed to stress showed cell type-specific transcriptional changes in glia, neurons, and vascular cells [34]. The authors also observed altered protein phosphorylation associated with stress, which completely reversed in 4 h in mice. However, an investigation of protein translation indicated that only a subset of transcripts becomes actively translated, varying among different cell types and subregions of the hippocampus [34].

Interestingly, scRNA-seq results have also provided evidence suggesting that not all cell types are easily characterized and clustered through unique specific markers in stress disorders. Transcriptional characterization of PACAP-expressing neurons in the lateral habenula failed to fit these neurons to a single and distinct subcluster, rather, they were disaggregated amongst other cell types in the scRNA-seq data [35,36]. However, chemogenic activation of these neurons in mice during behavioral tests showed disrupted fear memory and anxiety-association events that alleviated stressful situations [36], highlighting the need for a molecular strategy to isolate these neurons and evaluate their function in specific brain regions.

The majority of the current findings demonstrate associations between altered transcript production in specific cell types and the development of an aversive effect in behavior (i.e., depression and/or social avoidance) after conditioned stress exposure. These publications reveal powerful insights on relevant cell targets and the need to broaden brain regions being investigated in MDD studies. Single-cell and single-nuclei RNA-seq findings, mentioned in this manuscript, for MDD and the following psychiatric disorders are highlighted in Table 1.

## 3. Neurodegeneration: Parkinson’s and Alzheimer’s Diseases

Parkinson’s (PD) and Alzheimer’s (AD) diseases are neurodegenerative disorders that usually present with late-onset, affecting people in advanced age (>60 years old) and causing cognitive impairment among other clinical features [60]. While externally presenting as a movement disorder featuring resting tremors and rigidity, PD is known for the presence of Lewy bodies, starting at the dorsal motor nuclei of the vagus and spreading through the brain, followed by degeneration of dopaminergic neurons in the substantia nigra [60]. miRNA analysis from the blood and brains from PD subjects revealed known and novel biomarkers implicated in PD development and progression, such as axon guidance, TGF-β signaling pathway, ubiquitin-mediated proteolysis, and endocytosis [61]. Besides miRNA changes in PD-associated tissues, highly specific time-dependent intronic transcriptional changes were observed in blood RNA-seq but not yet tested in the brain [62].

Knowing that the loss of dopaminergic neurons is associated with the clinical outcome of PD, snRNA-seq helped to reveal the heterogeneity of dopaminergic neurons and distinguish temporal expression and trajectory profiles observed in midbrain neurons [37] and mesencephalic and subthalamic nucleus neurons [38] of mice. In vitro snRNA-seq performed in human iPSC-derived dopamine neurons carrying the PD-associated mutation SNCA-A53T showed transcriptional effects in different cell clusters and genes involved in cholesterol biosynthesis, glycolysis, and synaptic signaling pathways during oxidative stress [39].

Overall, snRNA-seq demonstrated an independent association of PD with transcriptomic changes in neurons (but not microglia) from the substantia nigra in human samples [40] and OL mainly from the medulla, midbrain, pons, thalamus, and spinal cord from one mouse brain dataset [41], regions that are all impacted by PD. In addition to cholinergic and monoaminergic neurons in the brain, enteric neurons were also associated with PD, reinforcing the gut-brain association with the disease through evidence of altered biological pathways in enteric dopaminergic neurons and glial components [40,41,63].

AD is observed after increased detection of β-amyloid (β-amyloid) plaques and deposition of neurofibrillary tangles and is clinically characterized by dementia with slow progress and variable evolution of symptoms [60]. Previous bulk RNA-seq and differential expression analysis studies highlighted features observed in AD progression: altered expression in microglia and inflammatory pathways in late-onset AD subjects, as seen with upregulation of *PLCG2*, a gene associated with cellular signal transmission in immune cells and microglia [64]; downregulation of DEGs associated with cell growth, proliferation, inflammation, and immune response, varying between three different pathological patterns for AD (typical neurofibrillary tangle pathology, hippocampal sparing AD, and limbic predominant AD) [65,66]; and the consistency and conservation of expression patterns in gene regulatory modules, where many upregulated AD-associated human genes were also upregulated in homologous mouse genes [67].

Additionally, snRNA-seq also revealed associations between disease progression and different transcriptomic profiles, as observed previously in bulk RNA-seq data. Individuals with no or very low β-amyloid burden (i.e., early stages of disease progression) presented cell type-specific differences in transcription in the PFC compared to individuals with high levels of β-amyloid (i.e., later disease stages) [42,43]. The main differences were observed in both excitatory and inhibitory neurons and glial cell types, mainly an OL subset marked by *CRYAB* [42]. Pathogenesis of the disease was further associated with somatostatin (SST) inhibitory interneuron loss and preservation of intratelencephalic-projecting pyramidal cells was associated with a slower rate of cognitive decline in human postmortem AD samples [44]. Both cell types showed diminished relative cell type proportions in AD samples compared to controls, and notably, decreased expression of SST interneuron genes was detected [44].

Investigations of PD and AD represent social and general interest, and further snRNA-seq will be able to explore more brain regions and cell types affected during the progression of both diseases, aiming for potential treatment targets. For example, brain snRNA-seq combined with serum proteome investigation recently revealed ten promising molecules as novel biomarkers for AD through the integration of DEGs and differentially expressed protein data [68], indicating encouraging future treatment directions.

## 4. Schizophrenia

Schizophrenia is a chronic psychotic syndrome, characterized by impairment in the perception of reality, with genetic and environmental influence [69]. Transcriptomic studies in human samples using bulk RNA-seq showed altered DEGs associated with neurotransmission, pre-synaptic function, neural development, and inflammation response [70]. Remarkably, energy metabolism and blood coagulation pathways, associated with mitochondrial functions, were implicated in the pathogenesis of schizophrenia [70]. Notably, Zhang et al. [71] identified one gene module, containing 89 genes, to be associated with abnormal psychomotor behavior in the disease in a bulk RNA-seq study of blood samples from people with schizophrenia. Dysregulation of genes associated with immune system response was further observed in blood [71,72] and in the amygdala tissue [73] of people with schizophrenia.

Similarly, snRNA-seq also provided evidence of the association between schizophrenia and inflammation. Using snRNA-seq in the midbrain from human postmortem tissue, schizophrenia-related genes indicated altered expression in BBB cells of the midbrain, which could increase inflammation [45]. The same study also showed transcriptional contribution to schizophrenia pathophysiology was limited to ependymal cells and pericytes, although relevant endothelial and astrocytes subpopulations were identified [45]. Altered gene regulatory pathways in schizophrenia were assessed more broadly through snRNA-seq studies in human fetal brains, which revealed that genes enriched for common risk alleles for schizophrenia presented high expression in neuronal populations of the frontal cortex, interneurons in the ganglionic eminence, and glutamatergic neuron populations of the hippocampus [46].

With the investigation of combined datasets from different approaches, many features associated with the cell dysregulations observed in schizophrenia were revealed. For example, through bulk expression datasets combined with multiple cell type-specific gene markers defined by single-cell data, Toker et al. (2018) reported that parvalbumin-expressing (PVALB) interneurons and cortical astrocytes displayed altered expression profiles in datasets from bulk tissue transcriptomics from people with schizophrenia [47]. Aggregation of human snRNA-seq data with GWAS data from multiple cortical and subcortical structures (neocortex, hippocampus, hypothalamus, striatum, and midbrain samples) showed enriched molecular pathways connecting schizophrenia-associated genes to hippocampal cornu ammonis pyramidal cells, neocortical somatosensory pyramidal cells (cortical layers 2/3, 4, 5, and 6), cortical interneurons, and striatal medium spiny neurons (MSNs) [22].

Overwhelming evidence also suggests that MSNs can influence the mechanisms of action of antipsychotic drugs used for schizophrenia treatment. D1R- and D2R-expressing MSNs from mouse striatum revealed significant differential gene expression when animals were treated with typical (haloperidol) or atypical (olanzapine) antipsychotics. Haloperidol had a higher impact in D2R-expressing MSNs, while olanzapine had similar effects on D1R- and D2R-expressing MSNs [74]. This evidence reveals that the pathology of schizophrenia and many other psychiatric diseases is broadly diffused across many brain regions and associated with highly heterogeneous cellular stratification. Ongoing snRNA-seq studies comparing brain tissues from individuals with schizophrenia and controls will provide deeper insight into the pathophysiology of the disorder and potential treatment avenues.

## 5. Alcohol Use Disorder

Substance use disorders stem from an unbalanced brain reward system directly associated with an individual’s motivated behavior. Several anatomically-defined regions in the brain, especially the ventral tegmental area, nucleus accumbens (NAc), and PFC, are associated with motivational states and addiction [75,76]. Recently, cellular profiling of these regions was performed in human and rodent samples [77,78,79], highlighting specific cell types involved in increased motivation [14,79].

One of the most common substances of abuse is alcohol. Alcohol dependence has been studied using many molecular approaches. The functional impact of specific genetic variants (*SPI1*, *MAPT*, *FUT2*), assessed through GWAS and cell type enrichment analysis data from the human brain, was associated with increased alcohol consumption and drinks per week [80]. Bulk RNA-seq revealed altered gene expression in the PFC and striatum in brain samples of alcohol-dependent human subjects [81,82] and in the rat’s central amygdala [83].

With single-cell transcriptome approaches, cell type-specific association with alcohol abuse was profiled [48,49,50]. In the PFC postmortem tissue from alcohol-dependent humans, astrocytes were the predominant cell type presenting altered transcriptomes, and DEGs were associated with neuroinflammation and apoptosis in response to chronic alcohol abuse [48]. In single cells from rats, obtained through laser microdissection and evaluated with microfluidic RT-qPCR, inflammatory gene clusters were also reported in the analysis of neurons stained for tyrosine hydroxylase and microglia of the nucleus tractus solitarius in alcohol-dependent rats [49]. This resulting phenotype seemed to be normalized after ~7 days of withdrawal, indicating the brain’s resilience to chronic alcohol exposure [49].

In the context of acute alcohol withdrawal, snRNA-seq revealed that one particular subcluster of the GABAergic neurons in rats, the protein kinase C delta (PKCδ—Prkcd) expressing neurons, had the largest number of DEGs, illustrating that those cells were more sensitive to the effects of acute alcohol withdrawal [50]. Additionally, higher frequencies of DEGs were reported in the central amygdala astrocytes and GABAergic neurons of rats [50,83]. SnRNA-seq has helped to better understand the molecular and cellular basis of alcohol abuse, but the connection between genetic and environmental factors that increase susceptibility and likelihood to abuse still needs to be demystified.

## 6. Opioid Use Disorder

The opioid epidemic has had considerable socioeconomic impacts in the United States and a recent exploratory study demonstrated an increased risk of other countries developing an opioid abuse epidemic [84]. The introduction of opioids in an individual’s life may start through a prescription for pain relief but can quickly escalate to abuse and addictive behavior in susceptible individuals [84].

Recent studies investigating gene expression through bulk RNA-seq approaches in multiple brain regions from rodents that became dependent or resilient in a heroin administration model identified transcriptome differences between the two phenotypes in the NAc and medial PFC (mPFC) [85,86]. In the rat NAc, genes associated with immunity, neuronal stimulation and outgrowth, and learning and behavior, were altered [85], and in the rat mPFC, genes related to schizophrenia, neuronal signaling, synaptic plasticity, RAS signaling pathway, and dendritic arborization were transcriptionally changed [86]. In addition to these variations, sex-specific impacts were also evident after repeated morphine administration, where DEGs in bulk RNA-seq were enriched for inflammatory pathways in male rats and depression in female rats [87].

Combined RNA-seq with imaging techniques showed PVALB- and SST-interneuron inputs and inhibitory transmission to pyramidal neurons enhanced reward after morphine use through the MOR and DOR pathways, respectively, in the prelimbic cortex of mice [88]. Also through RNA-seq, it was observed that transcription of genes associated with reward, such as *Pomc*, *Htr2a*, *Htr7*, *Galr1*, and *Glra1* genes, was altered in the striatum of mice after chronic oxycodone exposure, and the expression levels of some genes were correlated with the amount of substance taken [89]. This correlation between intake and altered gene expression is particularly concerning because opioid abuse was shown to affect miRNA signaling and neuronal circuit development after in utero exposure in rats with the consequent reduction in dendritic spine density and synapto-dendritic deterioration [90]. Oxycodone self-administration in mice also affected axon guidance and synaptogenesis genes in the NAc and caudate putamen [91]. Effects on the neurodevelopment of the offspring may, therefore, worsen with increased exposure.

Another concern about opioids is their effect on the immune system. Using scRNA-seq, opioids, particularly oxycodone, induced type I interferon signaling pathways in different cell types, like neurons and astrocytes, regulated by STAT1 transcription factor, in iPSC-derived forebrain organoids from subjects with opioid use disorder (OUD) [51]. The downregulation of interferon-stimulated genes has negative systemic consequences for the modulation of immune cells of PBMCs from opioid-dependent human subjects [52]. Single-cells collected from the central amygdala also showed that opioid withdrawal induced increased gene expression in the astrocytes in morphine-dependent rats, driving neuroinflammation through the upregulation of proinflammatory cytokines [53]. Interestingly, suppression of species associated with anti-inflammatory functions in the gut microflora was also observed during withdrawal [53]. Both changes were associated with anxiety-like behavior, similar to that of individuals during opioid withdrawal, and can contribute to drug-seeking behavior and negative reinforcement [53,92].

Non-neuronal cells, including oligodendrocytes and astrocytes, responding to morphine exposure in mice [6], and microglia markers in humans [93] were also previously reported or validated in snRNA-seq studies as significantly affected in OUD. A single intraperitoneal injection of morphine in mice led to the detection of oligodendrocyte-specific morphine-dependent differential gene expression (*Cdkn1a*, *Phactr3*, and *Sgk1*) in myelin-forming OL and mature OL in the NAc [6]. Our previous work also demonstrated transcriptional changes in OL between acute morphine exposure and repeated morphine self-administration in rats [54]. Besides OL, astrocytes and D1R- and D2R-expressing MSNs also had altered transcriptomes in the rat NAc [54]. Importantly, this work differentiated between cell type-specific transcriptional alterations that were associated with acute opioid exposure, repeated opioid self-administration, and the act of volitional opioid taking. Taken together, the first neural cell-type-specific dissection of striatal populations that are associated with volitional opioid administration is presented [54]. Extension of these findings to opioid withdraw and reinstatement, as a model of relapse, will expand our understanding of the molecular mechanisms that promote relapse during acute and chronic withdraw. Additionally, incorporation of epigenetic and proteomic approaches would expand our understanding of the complexity of the neural mechanisms of OUD.

## 7. Psychostimulants (Cocaine, Amphetamine, Nicotine)

The continuous use of psychoactive substances represents a public health concern due to addictive and adverse effects, potentially resulting in neurological damage [94]. In *Drosophila melanogaster*, a single exposure to cocaine led to sex-specific transcriptional responses, and among the most highly altered genes, 69% corresponded to human orthologs [95], suggesting potential translational effects in humans. In rodents, post-transcriptional changes in miRNA were reported after methamphetamine [96,97] and cocaine exposure [98]. Overall changes in mRNA expression related to axonal growth, neural plasticity, pre- and postsynaptic signaling, neurogenesis, and memory and learning skills, were observed after amphetamine (AMPH) treatment in overactive mice that became less hyperactive and comparable to the control group after AMPH introduction [99,100].

Using unaltered postnatal conditions or induced cocaine self-administration conditions, snRNA-seq revealed that distinct molecular cell types in the PFC of mice showed remarkable associations with different cortical layers (e.g., closely related excitatory neuronal subtypes were grouped in the same cortical layer) [55]. In addition, using snRNA-seq, DEGs also seemed to be more evident in D1R-expressing MSNs in the rat NAc after cocaine, which contained dopamine-responsive genes with a robust transcriptional response and overlapping transcriptional pathways with D2R-expressing MSNs [56].

The responses of different MSN subclusters in snRNA-seq have illustrated the cell type-specific effects of the use of psychoactive substances. A dichotomous response between D1R- and D2R-expressing NAc MSNs was evidenced through calcium imaging during repeated exposure to cocaine in mice, leading to increased activity of D1R-expressing neurons and decreased activity of D2R-expressing neurons registered during locomotor sensitization [101]. Complementing these findings, Chen et al. (2021) identified multiple subtypes of MSNs, using single-cell resolution with specific spatial localization, showing different roles for subtypes between the mouse NAc core and shell across the AP axis (e.g., Oprk1 was more abundant in D1R-expressing neurons clusters, while Oprd1 was restricted to D2R- expressing neurons), suggesting complex cellular interaction during the drug abuse process [102]. Besides MSNs, other GABAergic cells in the NAc are known to be affected by psychostimulant exposure in mice [102]. AMPH intake altered the expression of PVALB-expressing interneurons in mice, which was reflected in increased open-field locomotor activity, upregulation of genes that affect synapse structure, function, and excitability, and downregulation of general metabolic and biosynthetic pathways [57]. By isolating nuclei through INTACT (Isolation of Nuclei Tagged in Specific Cell Types), followed by bulk RNA-seq, two genes, *Cntnap4*, responsible for neurotransmission in GABAergic neurons, and *Acan*, an organizer of postsynaptic protein complexes, were seen to be induced by AMPH in PVALB-expressing interneurons in transgenic mice, despite no substantial changes reported in chromatin accessibility [57].

Nicotine is another psychostimulant that is known for addictive features and toxicity in smokers and people exposed to second-hand smoke [103]. miRNA and mRNA alterations, observed with RNA-seq, after withdrawal from chronic nicotine exposure, included downregulation of *Pfn2* in male mice [104]. Knockdown of this gene in the interpeduncular nucleus from the habenulo–interpeduncular axis causes anxiety-like symptoms in mice, similar to those of humans [104]. Using microarray analysis and PCR in human embryonic stem cells, nicotine treatment was shown to mainly affect stem cell differentiation pathways, with delayed cell differentiation and impacted embryonic germ layer development [105].

The impact of nicotine on human embryonic stem cell development was also addressed through snRNA-seq. He et al. (2020) showed that nicotine decreased cardiac progenitor lineages, mesodermal and neural crest cells, and inhibited cardiac-specific transcript factors during cardiac cell differentiation [58]. Nicotine also impacted cell-to-cell communication, decreased cell survival, increased reactive oxygen species generation, and affected the phases of cell cycling in embryoid bodies, causing abnormal signaling in cardiomyocytes [59]. Despite these cell type-specific effects and prior findings with other psychostimulants, single-cell techniques have not yet been used to study gene expression in individuals who smoke or use electronic cigarettes.

## 8. Future Directions

As shown in this review, there is considerable evidence supporting the need for increasing the number of snRNA-seq studies of psychiatric disease to improve the consistency and reliability in uncovering cell-specific gene expression features and revealing commonly shared and unique biological pathways [106]. However, some features and limitations must be considered moving forward, including a focus on increasing the number of cells/nuclei being analyzed, which will require more complex analysis capacity and development of tools for the inclusion of multiomic modalities and spatial resolution.

Currently, most studies rely upon a droplet-based approach to allow for the analysis of 10,000 cells in satisfactory conditions. Upcoming technological improvements will hopefully increase the number of cells and nuclei being analyzed per sample, especially for tissues constituted by heterogeneous populations of cells, like the brain. Barcoding strategies could also provide ultra-high-throughput through single-cell combinatorial fluidic indexing to enable the sequencing of millions of individual cells in one massive overloading [107]. A higher number of cells and regions being investigated will help to unravel complex systems and diseases and detect rare cell types. Higher throughput of cells will also provide a deeper understanding of sex-specific contributions to global or localized gene expression changes [108,109].

However, with a higher number of cells being analyzed, the need for more analytical tools arises to account for data interpretation and organization of information in an accessible way for the scientific community to address the missing gaps in research [12,13]. Generating more data in the future should also be tied with the opportunity and need to reanalyze and integrate previous data, especially with resolution improvement and the establishment of better cell type markers [18]. It is also worth mentioning that multiple datasets require the ability to perform cross-species comparison to translate single-cell findings from animal models to human data, using computational methods for orthological conversion and biological pathways comparison [110].

Having a deeper comprehension of gene expression across the brain will unravel many of the current molecular questions but not all of them. Therefore, the need for different single-cell omics emerges to investigate relationships at the chromatin level and also analyze epigenetic regulatory networks to identify correlations between the transcriptome with genome-wide DNA methylation [12]. Many epigenomic strategies have been developed recently: nano-body-based scCUT&Tag (Cleavage Under Targets and Tagmentation), which combines the molecular principles of the tethering method of specific sites using assigned nanobodies, antibodies, and Tn5 transposase and single-cell data to determine cell identity, transcription, and regulatory factors and histone modifications [111]; spatial-CUT&Tag, which uses an approach that identifies spatially resolved chromatin configuration [112]; NTT-seq (Nanobody-Tethered Transposition), which is capable of measuring multiple histone modifications and protein-DNA binding sites alongside the standard ATAC-seq method [113]; ASAP-seq, which can profile chromatin and protein level simultaneously [12].

Many other techniques are being updated and optimized to resolve transcriptional questions and produce multiomic data. The goal of the next single-cell assays should be focused on gene expression characterization combined with the spatial distribution of brain cells in multiple different psychiatric disease contexts, and recent papers are already addressing these gaps in the field [114]. Spatial single-cell RNA-seq will improve the resolution of transcriptomic findings and the development of targeted treatments in molecularly complex tissues, such as the brain. Generating multimodal data can provide the characterization of multiple genetic and epigenetic features from the same cell or nuclei to address biological heterogeneity, temporal evolution to disease state and spatial details for targeted treatments. Additionally, the need for the development of robust single-cell proteomic tools for addition into multiomics pipelines cannot be overstated, as they would allow for direct observation of the downstream effects of transcriptomic and epigenetic changes.

## 9. Conclusions

Given the complexity of psychiatric disorders, the characterization of each disease and unraveling of the underlying molecular features cannot be accomplished through a focus on only a few genes or regions. The neurobiological changes that occur in psychiatric disease and substance abuse have complex intra-organ and systemic interactions, polygenic features, and involvement of many different molecular pathways, necessitating circuit-specific and cell type-specific analysis. The push to better understand psychiatric disorders is an ongoing effort and snRNA-seq provides insights into transcriptional changes and cell composition. However, several questions remain: How can each molecular pathway involved in specific psychiatric disorders be better characterized? Will all relevant transcripts become actively translated into proteins and do the mRNA levels correspond to the levels of protein production? How will spatial transcriptomics illustrate the progression of disease stages? With the improvement and expansion of snRNA-seq studies, those questions and others may finally be answered.

## Figures and Tables

**Table 1 genes-14-00771-t001:** Single-cell or single-nuclei RNA-seq studies of psychiatric diseases.

Disease	Species/Type of Sample	Brain Area	Affected Cell Type	Reference
Major depressive disorder and stress	Human postmortem tissue	dlPFC	Immature oligodendrocyte precursor cells and excitatory neurons	Nagy et al., 2020 [32]
	Human postmortem tissue	dlPFC	Immune-oligodendrocytes and OPCs	Kokkosis et al., 2022 * [33]
	Mice postmortem tissue	Hippocampus	Glia, neurons, and vascular cells	Von Ziegler et al., 2022 [34]
	Mice postmortem tissue	Medial and lateral habenula	Habenula neuronal cell types and PACAP-expressing neurons	Hashikawa et al., 2020 [35] Levinstein et al., 2022 [36]
Parkinson’s disease	Mice postmortem tissue	Ventral midbrain	*Pitx3*-expressing neurons	Tiklova et al., 2019 [37]
	Mice postmortem tissue	Ventral mesencephalic and diencephalic region	Mesencephalic dopaminergic and subthalamic nucleus neurons	Kee et al., 2017 [38]
	Human iPSC	-	Dopaminergic neurons	Fernandes et al., 2020 [39]
	Human postmortem tissue	Substantia nigra	Dopaminergic neurons	Agarwal et al., 2020 [40]
	Mouse brain	Medulla, midbrain, pons, thalamus, and spinal cord	Oligodendrocytes and cholinergic, monoaminergic, and enteric neurons	Bryois et al., 2020 [41]
Alzheimer’s disease	Human postmortem tissue	PFC	Excitatory and inhibitory neurons and oligodendrocytes	Mathys et al., 2019 [42]
	Human postmortem tissue	Occipital cortex and occipitotemporal cortex	Microglia	Gerrits et al., 2021 [43]
	Human postmortem tissue	Neocortical regions	Somatostatin inhibitory interneurons and intratelencephalic-projecting pyramidal cells	Consens et al., 2022 [44]
Schizophrenia	Human postmortem tissue	Midbrain	BBB cells, ependymal cells, and pericytes	Puvogel et al., 2022 [45]
	Human fetal brains	Frontal cortex, ganglionic eminence, hippocampus, thalamus, and cerebellum	Neurons in the frontal cortex, interneurons in the ganglionic eminence and glutamatergic neurons in the hippocampus	Cameron et al., 2022 [46]
	Human postmortem tissue	-	Parvalbumin-expressing interneurons and cortical astrocytes	Toker et al., 2018 ** [47]
	Human postmortem tissue	-	CA1 pyramidal cells, neocortical somatosensory pyramidal cells, cortical interneurons, and striatal MSNs	Skene et al., 2018 ** [22]
Alcohol use disorder	Human postmortem tissue	PFC	Astrocytes, oligodendrocytes, and microglia	Brenner et al., 2020 [48]
	Rats postmortem tissue	NTS	Neurons and microglia	O’Sullivan et al., 2022 [49]
	Rats postmortem tissue	Central amygdala	Astrocytes, GABAergic neurons and protein kinase C delta-expressing neurons	Dilly et al., 2022 [50]
OUD—oxycodone and buprenorphine	Human iPSC-derived organoids from PBMCs	-	Neurons and glial cells	Ho et al., 2022 [51]
OUD—morphine	Human PBMCs	-	Immune cells	Karagiannis et al., 2020 [52]
	Rats postmortem tissue	Central amygdala	Neurons, microglia, and astrocytes	O’Sullivan et al., 2019 [53]
	Mice postmortem tissue	NAc	Oligodendrocytes and astrocytes	Avey et al., 2018 [6]
	Rats postmortem tissue	NAc	Oligodendrocytes, astrocytes, and D1R- and D2R-expressing MSNs	Reiner et al., 2022 [54]
Psychostimulants—cocaine	Mice postmortem tissue	PFC	Neuron subtypes	Bhattacherjee et al., 2019 [55]
	Rats postmortem tissue	NAc	D1R-expressing MSNs	Savell et al., 2020 [56]
Psychostimulants—amphetamine	Mice postmortem tissue	NAc	PVALB-expressing interneurons	Gallegos et al., 2022 [57]
Psychostimulants—nicotine	Human embryonic stem cells	-	Cardiac progenitor lineages, mesodermal and neural crest cells	He et al., 2020 [58]
	Human embryonic stem cells	-	Cardiomyocytes and neurons	Guo et al., 2019 [59]

Legend: BBB—blood–brain barrier; dlPFC—dorsolateral prefrontal cortex; MSN—medium spiny neurons; NAc—nucleus accumbens; NTS—nucleus tractus solitarius; OPC—oligodendrocyte precursor cells; OUD—opioid use disorder; PFC—prefrontal cortex. * Utilized Nagy et al., 2020 [32] publicly available dataset. ** Based on available datasets to connect transcriptomic data to specific cellular types.

## Data Availability

No new data were created or analyzed in this study. Data sharing is not applicable to this article.

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
