# Peer review of "Unraveling Psychiatric Disorders through Neural Single-Cell Transcriptomics Approaches"

_genes, 2023, doi:10.3390/genes14030771_

Round 1

Reviewer 1 Report

Chehimi at al. provide a nice summary and review of single-cell transcriptomic studies encompassing a variety of psychiatric disorders.  They note studies where single-cell approaches uncovered additional interpretable results than were identified by previous methods. 

One conceptual issue that comes up in many discussions of snRNAseq results, particularly in brain, is the interaction between gene expression and cell identification.  We usually use gene expression patterns to classify cell types.  But then we compare disease to control for differences in gene expression.  We should be careful to note that these changes may be sufficient to alter our definitions of cell types or subtypes.  It’s only a suggestion, but the authors might consider some mention of this issue to be more critical.

Only a few issues were detected that could improve the impact of this review. 

1.       One major concern was that the point of the review is to show how single-cell methods produce more interpretable results in models of psychiatric disorders, but it isn’t always clear which results were uniquely found in single-cell models.  The authors list results that probably are specific to single-cell studies, but they don’t make this point, and doing so would greatly reinforce their main thesis.  One example is the observation in SST neurons in AD (Line 178).  Was this first (or only) shown in snRNAseq?  If so, say it.

2.       The authors switch between human and rodent models, sometimes without adequately describing this.  One example is page 3, line 113.  The previous paragraph clearly described results from persons with MDD, but this paragraph described results following stress, which could be reversed after 4 hours.  Hopefully, this was in an animal model.  Another example is in the OUD section, where gene symbols appear to be from non-human samples but this is not made clear.

3.       Similarly, when used, animal models are not always adequately described.  Which mouse MDD model was used in line 113, for example.

4.       The authors often say “changes in transcription” when they mean changes in gene expression or steady-state mRNA levels.  Not all mRNA changes are due to transcription.

5.       Future Directions—don’t you want to mention spatial transcriptomics? 

Minor points:

1.       It appears that some abbreviations are described and then used only once.  Unless you use it repeatedly, it’s better not to introduce abbreviations.

2.       In the abstract, line 7.  Suggested addition underlined: “The development of single-cell and single-nucleus transcriptome technologies”?  Similarly in line 12, “importance of transcriptome analysis of neuronal cells….”

3.       In line 127, you say the “findings reported here” but the results were published elsewhere, so they really weren’t reported “here.”

4.       In the AUD section, reference 66 is used to give examples of DEG, but this reference studied m6A modification of RNAs, not DEG.

5.       Lines 173-177—this sentence isn’t wrong but it took me three tries to understand what you were trying to say.  Can you simplify?

6.       Line 261, when introducing OUD, the authors give mortality statistics.  It seems odd that this was the only case that included mortality/morbidity. 

Reviewer 2 Report

The paper "Unraveling psychiatric disorders through neural single-cell transcriptomics approaches" by Chehimi and colleagues is a review on the potential of a relatively new technology on unraveling the molecular features of extraordinarily complex pathologies. The authors show both general and practical understanding of the clinical and technical aspects of single-cell transcriptomics performed on brain samples, and provide a general overview on a very actual topic. While the writing is good and the manuscript is generally pleasant to read, I believe the paper currently suffers from several issues, listed below.

MAJOR POINTS

- The paper currently lack a global synthesis of the findings that have been found using the novel technology of single-cell on neural tissues. In other words, it lacks the element of "Unraveling psychiatric disorders" in the title. I would suggest the authors to perform a more in-depth analysis of the current available public single-cell neural data, to highlight common recurring patterns in psychiatric disorders (one example, is the ever-present upregulation of inflammatory pathways in microglia, found in Alzheimer's, Schizophrenia, et cetera; another one is the common patterns found in several addiction disorders). An example of a great previous review highlighting common patterns and the utility of single-cell transcriptomics for psychiatric disorders is https://www.nature.com/articles/s41380-021-01324-6.

- There are several missing psychiatric disorders which have publicly available scRNA-Seq or snRNA-Seq data, and/or could benefit from single-cell resolution in future studies, which have not been covered by the authors. One example is autism. Another example is neuroinflammation elicited by HIV infection (commonly referred to as "neuro-HIV",see for example https://www.ncbi.nlm.nih.gov/pmc/articles/PMC8613726/ by Corley and Farhadian, 2022). I believe the authors should add a paragraph on all the psychiatric disorders currently not covered.

- In connection to the previous points, I would highly suggest to provide the reader with either a summary table or a summary figure of their review, highlighting for example the brain areas, cell types, species, and disorders discussed in the review, with a particular focus on the combinations covered by existing public datasets.

- Line 29: beyond FACS and laser microdissection, also computational methods of deconvolution should be mentioned as tools to infer cell type-specific composition of bulk samples (e.g.  https://www.nature.com/articles/s41467-020-19015-1).

- Line 61: some more introduction on the currently existing methods to classify brain samples based on known cell markers should be written. For example, I would mention the collective consotirum efforts resulting in the Azimuth brain cell annotator https://azimuth.hubmapconsortium.org/

- Some of the references seem to be improperly placed or cited, so I would suggest the authors to do a thorough assessment of all the referenced work, specifically of the papers claiming to be performing single-cell studies. For example, reference 30, which authors claim to contain single-nucleus data (line 109) refers to a study that only implements the qRT-PCR as a quantitative transcriptomics technology. Moreover, the study is performed in mice, not in humans as claimed by the authors. Another example is reference 25, cited as an example of single-nucleus RNA-Seq PFC on line 110, but in fact is based on a collection of arrays and bulk RNA-Seq data. A third example is reference 33, which indeed shows single-cell data, but it's derivative of another previous work of the previous team (PMID 32272058); also referring to this, while the work of reference 33 is cited as rat-based (line 123), the actual single-cell data is based on mice. We admit we did not check every single reference in the paper, but I urge the authors to double-check all references, in particular those discussing single-cell data in psychiatric disorders (being the focus of the review).

- I would suggest the authors to expand the parts that are single-cell specific. Currently, every disorder-specific paragraph contains a brief introduction on the clinical and social aspect of the disorder, a long excursus on the existing molecular and histological knowledge on that disorder, and then an equally sized part on studies based on single-cell data. In general, I would increase the focus on the novel findings, with, as I suggested previously, an effort in generating a synthesis of what is known, within and across disorders, thanks to single-cell gene expression quantification.

- Line 375, referring to the previous point, in my readings of single-cell studies on brain disorders, the classification of cell types is also of paramount importance to understand the histologically-specific transcriptional phenomena. For example, neurons are "easily" distinguishable from other cell types (such as microglia, astrocytes, OPCs), but there is more debate within the neuronal populations, with for example a basic distinction between GABAergic and glutamatergic neurons, and more nuanced categorizations, based on layers, projection types, and the expression of specific markers. Also, hybrid populations exist (e.g. neurons with both GABAergic and glutamatergic markers). The authors should mention that better classification of single-cells would also provide a more cell type-detailed focus on the molecular basis of psychiatric disorders, and that reanalysis of existing datasets in view of the most recent findings on cell markers (e.g. Azimuth) could be a worthy endeavor to improve the resolution of our understanding.

- The species in which mentioned studies are performed is often, but not always specified (e.g., line 174). The authors should go through the manuscript and expand on this, by specificying whether each of the cited findings are in human or in a model.

- Line 391: in the discussion, the authors correctly mention that scRNA-seq (and snRNA-seq) alone cannot provide answers to all current molecular questions, mentioning for example the necessity to pair transcriptomics with sc-ATAC-Seq and ASAP-seq. I would however also explicitly mention the other needed revolution, that of spatial transcriptomics, which will provide not only single-cell expression, but also the 3D localization of that cell in the tissue of origin (currently, this is briefly formulated as a question in line 425).

- Future directions are very generic, so far that they could be applied to any tissue and pathology. The authors should expand on why the brain represents a unique challenge and opportunity for the signel cell/single nucleus RNA-Seq technology, given its absolutely fantastic histological and structural complexity.

MINOR POINTS

- Line 139: "miRNA" in theory should remain in lower case even at the beginning of a sentence.

- Scientific names (such as Drosophila melanogaster) should be in italics.

- Line 109: it's not "loss of CLDN5", but rather "decrease of CLDN5", since the expression is decreased to 50% in the cited work.

- The authors mention the importance of animal models for brain studies. However, it is not always obvious how to convert human single-cell data to animal model data. The authors should mention wither the necessity and the potential drawbacks of gene id translation, or even a few tools for orthological conversion, such as Homologene, DIOPT, et cetera, which are commonly employed to do a gene-by-gene translation between species. Other types of analysis may skip the gene-by-gene conversion, and rather translate genes directly into pathways (which are, broadly speaking, more "conserved" than individual genes in their activity in single, physiological and pathological, cells).

Round 2

Reviewer 2 Report

The points raised by both reviewers were largely overlapping, and the authors did their best in replying to them. As a result, the manuscript is now more robust and lacks the numerous clerical errors in citations of the first version.

Only one point remains: in the new Table 1, the authors use as reference the generic form "Author et al., year", which is hard to find using the current reference scheme. The authors should use the same reference system as in the rest of the publication, so in the form [1] [2] [3].

Author Response

We thank the reviewer for the observation. We have included the numbered references in the table, along with the first author's last name and year.